# How Humoral Response and Side Effects Depend on the Type of Vaccine and Past SARS-CoV-2 Infection

**DOI:** 10.3390/vaccines10071042

**Published:** 2022-06-29

**Authors:** Monika Stępień, Małgorzata Zalewska, Brygida Knysz, Natalia Świątoniowska-Lonc, Beata Jankowska-Polańska, Łukasz Łaczmański, Agnieszka Piwowar, Amadeusz Kuźniarski

**Affiliations:** 1Department of Infectious Diseases, Liver Diseases and Acquired Immune Deficiencies, Wroclaw Medical University, 50-367 Wroclaw, Poland; zalewskamalgorzata9@gmail.com (M.Z.); brygidaknysz@gmail.com (B.K.); 2Center for Research and Innovation, 4th Military Teaching Hospital, 50-981 Wroclaw, Poland; natalia.swiat@o2.pl (N.Ś.-L.); bianko@poczta.onet.pl (B.J.-P.); 3Hirszfeld Institute of Immunology and Experimental Therapy PAS, 53-114 Wroclaw, Poland; lukasz.laczmanski@hirszfeld.pl; 4Department of Toxicology, Wroclaw Medical University, 50-367 Wroclaw, Poland; agnieszka.piwowar@umw.edu.pl; 5Department of Dental Prosthetics, Wroclaw Medical University, 50-367 Wroclaw, Poland; optymalizacja@aps.wroclaw.pl

**Keywords:** SARS-CoV-2 vaccines, COVID-19, humoral immune response, adverse effects of vaccines, vaccination program

## Abstract

Since the end of December 2020, it has been possible to vaccinate against COVID-19. Our aim was to evaluate and compare the effectiveness of the vaccines available at the time of the mass vaccination program in Poland and also to look into the most common adverse side effects. Patients’ anti-SARS-CoV-2 antibodies levels were checked before vaccination and after the first and after the second/last dose by the anti-SARS-CoV-2 QuantiVac ELISA (IgG) (EUROIMMUN MedicinischeLabordiagnostica AG; Luebeck; Germany) test. Before each blood collection, all patients filled out a questionnaire regarding experienced side effects. We observed that 100% of patients responded to the vaccinations. After the first dose, convalescents had much higher levels of anti-SARS-CoV-2 antibodies than naive patients, although after the second dose, 61 out of 162 convalescents (37.7%) had lower results than before. The comparison of immunological responses in the convalescents group after the first dose and in the naive group after the second dose showed that convalescents had higher antibody titers, which may suggest the possibility of changing the vaccination schedule for convalescents. The highest antibody titers after both the first and second doses were observed after Moderna shots. Fever was identified as a significant factor regarding higher levels of antibodies after the first and second doses of the vaccine.

## 1. Introduction

Since the end of December 2020, it has been possible to start vaccination against COVID-19. Safe and effective vaccines were introduced at least one year after the beginning of the COVID-19 pandemic and have helped to control it in many regions of the world. There are five vaccines used in Poland which have been approved for use by EMA. Two of them are mRNA vaccines—Pfizer/BioNTech and Moderna (New York, NY, USA)—and the other two are based on the adenoviral vectors by Astra Zeneca and Johnson&Johnson and the last one, the recently introduced Nuvaxovid (Novavax), is a protein subunit vaccine.

Many studies have started to evaluate the efficacy, safety and tolerability of the different types of vaccines. It turns out that COVID-19 vaccines provide sufficient protection against serious illness and death, although the efficacy can be lower in terms of new variants such as Delta and Omicron—especially sub-lineages BA.4 and BA.5, which are now classified as VOC [1]. On the other hand, vaccination can prevent to some extent transmission of the virus to others. However, waning of the immune response and appearance of ‘breakthrough infections’ in fully vaccinated patients can be seen. It can happen more often in older people with underlying medical conditions or who have high-risk exposures to SARS-CoV-2 [2,3].

It has been established that additional doses of COVID-19 mRNA (Pfizer or Moderna) vaccines should be used to improve decreased protection after the primary vaccination schedule, especially in terms of new VOCs. The CDC has approved booster doses: a first booster dose for everyone 5 years of age and older, and the second booster for adults 50 years of age and older and persons 12 years of age and older who are immunocompromised [4].

Still, there are not enough data on the size and duration of natural immunity and immunity induced by vaccination. It can depend on viral, host and demographic factors. There are reports indicating that convalescents achieve enough protection soon after one vaccine dose and that antibody titers after the first dose in convalescents and the second dose in patients with no previous infection are similar. However, data concerning long-term protective immunity in both convalescents and naïve patients require more studies [5,6,7,8].

Many individuals experienced side effects due to vaccination. Most often, they are mild to moderate and include fever, headache, fatigue, malaise, myalgia, chills, pain or redness at the injection site. Sometimes, side effects are more serious more serious. Unexpected side effects of COVID-19 vaccines such as allergic reactions, anaphylaxis, immune thrombocytopenia, cerebral sinus venous thrombosis or splanchnic vein thrombosis with antibodies to platelet factor 4 have been reported [9].

In mRNA COVID-19 vaccination, rare cases of myocarditis and pericarditis, mostly among males ages 12 through 39 years, have been reported [10,11].

Seroconversion panels based on serial tests of blood samples collected before and after each vaccine dose can demonstrate useful methods to control exposure to SARS-CoV-2 and vaccine effectiveness.

The aim of this study was to evaluate anti-SARS-CoV-2 antibody response after the first and the second doses of vaccines in a cohort of adult patients (previously infected or not infected) from the Lower Silesia Region, Poland, and to compare antibody titers after each vaccine dose in these two groups. We also looked for the most common adverse effects between different types of COVID-19 vaccines and possible SARS-CoV-2 infections during and after vaccinations.

## 2. Materials and Methods

The study received approval of the Bioethic Committee at Wroclaw Medical University (No 51/2021).

### 2.1. Study Group

The single-center study was performed at the Medical University of Wroclaw, Poland, between 20 February 2021 and 25 August 2021. The patients were recruited by an announcement in local media such as newspapers, television and the local hospital’s website. They filled in a contact form and were called or messaged by members of our team to confirm their willingness to be vaccinated against SARS-CoV-2, to take part in our study and to initially exclude any contraindications. Later, in person, the volunteers signed a questionnaire, where they had to provide information about exclusive diseases such as diabetes, any cancer within the last 5 years, chronic kidney, liver or lung diseases, AIDS or immunosuppression for any other reason. The patients registered for a specific date. The choice of the vaccine and the facility providing COVID-19 shots were random. One of the vaccines against COVID-19 registered at that time in the EU (Pfizer, Moderna (Cambridge, MA, USA), Astra Zeneca (Cambridge, UK) or Johnson & Johnson (New Brunswick, NJ, USA)) was subsequently used. Each patient was informed about the aim of the study. Patients were informed that they could withdraw their consent at any stage of the study and they signed informed consent. It was also necessary for the volunteers to sign in for the vaccination against SARS-CoV-2 before joining the study. In the first part of the study, we assessed anti-SARS-CoV-2 antibody levels before vaccination. Due to frequent changes in registration rules and vaccination dates and patients’ individual health contraindications at the moment, the interval between collecting blood samples for testing and the first dose of vaccine varied from 1 day to 6 weeks, usually approximately 1 week.

Later, we invited the patients for antibody level follow-ups after the first dose, precisely up to 7 days before the second vaccination and again 4–5 weeks after the second/last dose.

Inclusion criteria: Patients over 18 years of age who signed informed written consent to participate in the study and were willing to be vaccinated were included. The patients disclosed whether they passed SARS-CoV-2 infection and if it was confirmed with PCR or serological test.

Exclusion criteria: Those who suffered from diabetes, any cancer within the last 5 years, chronic kidney, liver or lung diseases, AIDS or immunosuppression for any other reason were excluded.

Before each blood sample was taken, at every visit, information on adverse events related to vaccination and general health conditions was collected. Questions in a self-administered questionnaire concerned general well-being, persistence of COVID-19 symptoms, adverse vaccine reactions, treatment administered, chronic diseases and allergic reactions to drugs, substances and foods. Patients were helped to complete the questionnaire by a team consisting of a doctor and a nurse.

According to the inclusion and exclusion criteria and questionnaire data, 298 patients, citizens of Lower Silesia region, Poland, mostly from Wrocław, aged 21–69 of both sexes were enrolled to the study. Originally, both groups had similar sizes, but 21 patients declaring themselves as naive, with no history of COVID-19 symptoms or laboratory confirmation of the infection, had antibodies present before vaccination. This subgroup of asymptomatic convalescents was separated from the naive group and transferred to the convalescents group. Based on final serological results obtained before vaccination in the whole study group, two groups of patients were established:Group I of 171 COVID-19-convalescent individuals with positive results on PCR, antigen or serological tests confirming the presence of IgG antibodies or other strong indication of past infection (e.g., loss of sense of smell after living with someone with confirmed infection), within 6 months prior to qualification for this study;Group II of 127 patients without evidence of previous SARS-CoV-2 infection and with 0 anti-SARS-CoV-2 antibodies before vaccination (naive patients).

During the study, the numbers of each group had been changing due to COVID-19 diagnoses after the first dose of vaccination, resigning from the study after the first or second test, admitting volunteers after the first dose if they presented individual IgG antibody results from before the shot. These factors resulted in the following number of participants tested in each phase of the study:

D0 (test before vaccination) = 298 participants (group I: 171, group II: 127);

D1 (test after 1st dose) = 286 (I: 163, II: 123);

D2 (test after 2nd/last dose) = 295 (I: 169, II: 126).

All individuals participated in the first part of the study, in which we assessed initial antibody titers (before vaccination). The manuscript titled Epidemiological and retrospective study in cohort qualified for anti-SARS-CoV-2 vaccination in the region of Lower Silesia, Poland was send to the Editor of a journal for publication. The majority of the study group received mRNA vaccination (Pfizer n = 191; Moderna n = 70), while others chose vector shots (Astra Zeneca n = 25, Johnson&Johnson n = 9). Since Johnson&Johnson full vaccination consists of only one dose, individuals vaccinated with it were tested just once and their results were compared with the results of the other patients after the second doses of other vaccines.

Patients were obliged to come for a visit before the first (D0) and the second dose of vaccine (D1) (up to 7 days before the second dose) and also 4–5 weeks after the second /last dose (D2). The intervals between D0 and D1 varied greatly due to different time periods between the first and second vaccines (originally, for Pfizer 21 days, Moderna 28 days, Astra Zeneca 3 months, but the recommendations changed with time).

### 2.2. The Procedure

Blood samples were taken before (D0) the first and second dose of vaccine (D1) (up to 7 days before second dose) and 4–5 weeks after the second/last dose (D2).

At each visit, the patients were tested for anti-SARS-CoV-2 IgG antibodies. Plasma samples were collected using heparin, centrifuged and stored in aliquots at −70 °C for later use. The anti-SARS-CoV-2 QuantiVac ELISA (IgG) (EUROIMMUN MedicinischeLabordiagnostica AG, Luebeck, Germany) was used for quantitative detection of anti-SARS-CoV-2 antibodies by means of a 6-point calibration curve.

In the quantitative enzyme-linked immunoabsorbant assay, the S1 domain of the spike protein of SARS-CoV-2 including the receptor binding domain (RBC) was used as an antigen. ELISA assay was performed and the results were evaluated as recommended by the manufacturer. Samples with absorbance higher than the absorbance of the highest standard (386 IU/mL) were diluted and retested. The final results were calculated by multiplication by a dilution factor. The assay was standardized against “First WHO International Standard for anti-SARS-CoV-2 immunoglobin” (NIBSC 20/136), so the quantitative results are given in standardized units: IU/mL (IU—international units) which are identical to BAU/mL (BAU—binding antibody units).

### 2.3. Statistical Analysis

For each parameter, mean, median (M), standard deviation (SD), range (min, max), lower and upper quartile (25Q, 75Q) were calculated. Statistical significance between means for independent groups was calculated by one-way analysis of variance (ANOVA), alternatively using the non-parametrical Mann–Whitney U test, when the variances in groups were heterogeneous (the homogeneity of variance was determined by the Levene’s test).

Statistical significance between frequencies was calculated by the chi-square test with corresponding degree of freedom df (df = (m − 1) ∗ (n − 1), where m—number of rows, n—number of columns). A *p* value of less than 0.05 was required to reject the null hypothesis. Statistical analysis was performed using EPIINFO Ver. 7.2.4.0 and Statistica Ver. 13.3. software packages.

The primary outcome was antibody response to 4 different vaccines at D1 and D2 in two groups of patients: convalescents and those with no evidence of previous SARS-CoV-2 infection. The secondary outcome was the frequency and the type of adverse effects following the first (D1) and the second (D2) dose of vaccine reported in the questionnaire.

## 3. Results

At the beginning of the project, 298 participants were divided into two groups—group I: 171 patients with previous SARS-CoV-2 infection (in the initial tests before vaccination, the median results for men was 164.7 IU/mL, for women 226.6 IU/mL) and group II: 127 naive patients (initial tests did not detect any antibodies). Table 1 presents the characteristics of the group of patients. This table in a modified version was also presented in the previous paper sent to another journal and is currently under review.

In this paper, we were focusing on participants that took part in second (D1) and third (D2) antibody tests. Therefore, total abundance on each step of the study slightly differs.

From group I, 150 patients had test-proven or highly possible SARS-CoV-2 infection between 1 October 2020 and 5 April 2021, of which 3 were hospitalized due to the severity of COVID-19 symptoms. Additionally, 21 people were found seropositive without any knowledge of previous infection.

The mean time between the diagnosis and 1st dose of vaccine totals 153.7 days, median 169 [IQR 128.0; 185.0].

The mean age of both groups was similar and there were no significant differences in terms of sex, with slight predominance of women in the groups. There were no differences between group I and group II regarding co-morbidities.

At the baseline, anti-SARS-CoV-2 IgG antibodies were found only in convalescent group I with a median number of 105.6 [38.4–198.4] IU/mL. Results obtained in men and women did not differ. In group II, no spike-antibodies were seen (0 IU/mL).

All participants received a COVID-19 vaccine (Pfizer-BioNTech—64.74% or Moderna 23.73%, AstraZeneca (Cambridge, UK) 8.47%, Johnson&Johnson 3.05%) and were tested up to 7 days before the second dose of two-dose vaccines and 4–5 weeks after the second (or last dose in those receiving the Johnson&Johnson vaccine). During this part of the study, mixing vaccines was not allowed, meaning the second shot had to be the same as the first one.

From all participants, we received information about only one SARS-CoV-2 infection after the first dose of the Moderna vaccine. This woman was vaccinated on 23 April 2021, and she received a positive RT-PCR result on 5 May 2021. The second dose was administered on 21 June 2021. Before vaccination, she was not infected and her initial antibody level from a sample taken on 21 April 2021 was 0 IU/mL. After infection, she did not match any of the two analyzed groups; therefore, she was excluded from further analysis.

Levels of anti-SARS-CoV-2 antibodies and increase of antibody levels were analyzed separately in each group and also compared between group I and group II after each vaccine dose.

### 3.1. Group I—Convalescents

In group I, 86.5% were initially seropositive (before vaccination). After the first dose (0–7 days before 2nd dose), 98.77% of 163 participants had IgG antibodies present in the blood samples; 2 people did not respond to the vaccine:patient 1 (group I)—male, age 27, symptomatic SARS-CoV-2 infection in the past, antibodies levels: D0 = 57.6; D1 = 0; D2 = 2860 IU/mLpatient 2 (group I)—female, age 39, symptomatic SARS-CoV-2 infection in the past, antibodies levels: D0 = 0; D1 = 0; D2 = 1318 IU/mL

In group I, a positive correlation was found between baseline (D0) and D1 antibody levels (*p* = 0.0352). The results after the first dose are presented in Table 2.

We did not find a similar corelation between baseline and D2 antibody levels (*p* = 0.135).

### 3.2. Group II—Naïve Patients

In group II, no anti-SARS-CoV-2 antibodies were present initially. After receiving the first vaccine dose, 98.37% of patients responded; two patients did not produce antibodies:patient 3 (group II)—male, age 61, antibodies levels: D0 = 0; D1 = 0; D2 = 1050 IU/mLpatient 4 (group II)—female, age 38, antibodies levels: D0 = 0; D1 = 0; D2 = 2464 IU/mL

In general, in both groups we observed a considerable increase of antibody levels after the first dose; in group I, there was a fifty-two-fold increase. Antibody levels were significantly higher titers after the first dose (D1) in group I compared to values obtained in group II (median group I: 5501.5; group II: 751.9 IU/mL) and after the second dose (D2) (median: group I: 5523.7; group II: 3625.6 IU/mL). For detailed information, see Table 3.

### 3.3. Lower Results after Second Dose of Vaccine

After the second dose, all participants reacted in an expected way and in 100%, samples taken 4–5 weeks after the full vaccine scheme, IgG antibodies against SARS-CoV-2 were found (including 4 people who did not respond to the first dose). In group I, we observed a slight increase in mean antibody titers (5501.5 vs. 5523.7 IU/mL), whereas in group II the increase was visibly higher (751.9 vs. 3625.6 IU/mL, which is an almost fivefold increase).

These differences in responding to the first and second doses in each group (group I presented a minimal increase in antibodies between D1 and D2, whereas group II presented continuous growth of antibody levels) led us to look deeper into the collected data. We were able to separate out from group I (convalescents) patients who had lower levels of antibodies after the second dose than after the first dose—61 out of 162 analyzed individuals (37.7%). We distinguished this subgroup with the letter ‘A’ and rest of the patients who had as-expected higher antibody titers after second dose with the letter ‘B’. A careful analysis of this topic showed that there was a positive correlation between a higher amount of antibodies after the first dose and the possibility of having a lower result after the second dose—the higher the antibody levels after the first dose, the more likely they would fall despite/after the second dose (*p* = 0.0001). We also discovered a possible influence of age on increased antibody production: younger participants were in subgroup (A) with lowering antibody results after the second dose (mean age 42.3 vs. 45.4; *p* = 0.0418). We did not observe a correlation between the analyzed changes and baseline antibody levels (*p* = 0.939) or results after the second dose (*p* = 0.0542). For more, see Table 4.

### 3.4. Antibody Titers after First Dose in Convalescents and after Second Dose in Naïve Patients

Another interesting observation concerns the antibody levels in group I after the first dose and group II after the second dose. Aware of the fact that previous SARS-CoV-2 infection is similar to immune priming and the first vaccine dose in group I can be compared to a booster, we analyzed antibody levels reached after the first dose of the vaccine in convalescent group I and after the second dose in naive group II.

Assuming antibody levels are a sufficient measurement method of immunological response, it may be concluded that convalescents had been more immunologically stimulated with one dose than naive patients even with two doses (*p* = 0.00003). In comparing group I after the first dose vs. group II after the second dose, the numbers were as presented: mean ± SD 5501.5 ± 4380.0 vs. 3625.6 ± 2568.9 IU/mL; M [IQR] 4736.0 [2676.9; 6144.0] vs. 3056.0 [2048.0; 4512.0].

### 3.5. Specific Vaccines

During the analysis of the collected data, a summary of antibody results after particular vaccines was prepared (see Table 5). Because of the previously explained factors, the number of participants at each stage of the study was slightly different. The group sizes were as follows: for Astra Zeneca D0 = 24, D1 = D2 = 25; Johnson&Johnson D0 = D1 = D2 = 9; Moderna D0 = D1 = D2 = 70; Pfizer D0 = D1 = 192, D2 = 191. The highest antibody titers after both the first and second doses were observed after Moderna shots.

None of the participants was infected with SARS-CoV-2 4 weeks after full vaccination.

Table 6 shows the relationship between the effect of the second dose of the vaccine depending on the formulation administered. We observe a statistically significant difference between the Astra Zeneca, Moderna and Pfizer preparations.

The below Figure 1 shows that Moderna induces the production of antibodies the most, similarly to Pfizer, but Astra Zeneca and Johnson&Johnson are the least active.

### 3.6. Adverse Effects

During the study, the adverse events after vaccinations were strictly reported. Most of them were mild symptoms lasting 1 or 2 days. We analyzed them separately regarding groups I and II and also first and second doses of vaccines. Reported side effects are presented in Figure 2 and Figure 3.

Some of the symptoms turned out to be more significant in relation to antibody production. In group I, we were able to identify increased temperature as a particular factor. We found a correlation between the presence of fever after the first dose with a higher antibody titer baseline (*p* = 0.0476), after the first dose (*p* = 0.0222) and second dose (*p* = 0.00026), and this was the only significant symptom after the first dose. Fever after the second dose was related to higher antibody levels after the first dose (*p* = 0.00027) and second dose (*p* = 0.00002). Weakness after the second dose was visibly associated with the immunological response after the first dose (*p* = 0.00000) and second dose (*p* = 0.00000). There is also a possible correlation between pain at the injection site after the second dose and antibody levels after the first dose (*p* = 0.0458) and second dose (*p* = 0.0161) and between muscle and joint pain after the second dose with antibody results after the first dose (*p* = 0.00613) and second dose (*p* = 0.0221).

Similar correlations were observed in group II regarding symptoms after the second dose: between fever and antibody titers after the first dose (*p* = 0.00001) and second dose (*p* = 0.00026); between pain at the injection site and antibody levels after the second dose (*p* = 0.00515); between weakness levels after the first (*p* = 0.0107) and second dose (*p* = 0.0114); and between muscle and joint pain and antibody titers after the first (*p* = 0.00815) and second dose (*p* = 00460).

We also arranged the adverse events with regard to specific vaccine types. The data are presented in Figure 4 and Figure 5.

## 4. Discussion

We reported the antibody response (IgG) after the first and the second doses of four different vaccines against COVID-19 authorized by the FDA and EMA in a cohort of adult patients from the Lower Silesia region in Poland reflecting the general population. These patients were divided into two groups: group I with previous SARS-CoV-2 infection and group II not infected before vaccination. Analysis of different parameters such as age, sex, time elapsed since natural infection that could influence humoral response and adverse side effects were explored. Although immunological defense is based on many different mechanisms, we believe that analyzing precise antibody titers, as a simple, minimally invasive, cheap and widely available method contributes strongly to the validation of the effectiveness of the COVID-19 vaccines. We compared the results between group I and group II in relation to their baseline antibody levels as well as antibody levels after the first dose in group I and second dose in group II. We also analyzed the group of patients with previous infection in whom antibody levels after the second dose were lower than after the first dose of vaccine.

We found that the presence of antibodies at baseline influenced the response to the vaccine after the first dose, but not after the second one. In general, previous SARS-CoV-2 infection resulted in significantly higher IgG anti-SARS-CoV-2 antibody levels compared with group II without evidence of previous infection. Our results are consistent with the data reported by others. B. Wolszczak-Biedrzycka reports that 8 months after two doses of the Pfizer vaccine among health workers, anti-SARS-CoV-2 S antibodies were still detectable and considerably higher in the group of convalescents compared to people vaccinated without a history of SARS-CoV-2 infection [12].

Regarding our results, Mendoza-Gonzales et al. found that antibody concentrations after the second dose of Pfizer-BioNTech were similar in both groups although significantly higher among convalescents after the first dose [13]. In our study, convalescents also presented higher antibody titers after the second dose. Paul R.Wratil et al. reported that convalescents developed a higher neutralization capacity against all SARS-CoV-2 variants of concern than naive individuals after vaccination, and that in naive individuals, the infection-neutralization capacity after the second vaccination was significantly lower than that of vaccinated convalescents [14].

It was observed that previous infection may be compared to the first vaccine dose because of the immune priming, and the first dose of vaccine would be similar to the booster. There is a question of whether the second dose in a basic vaccination schedule of an anti-COVID-19 vaccine in the group of patients previously infected is needed. There are a few reports indicating that previously acquired immunity due to infection with SARS-CoV-2 is connected to the level of antibody response to the first vaccine dose similar to that achieved in naive individuals after the second dose [8,15,16].

The results published by Jung et al. and Gaebler et al. show that individuals with past SARS-CoV-2 infection produce memory B and T cells that protect against re-infection even for 10 months [17,18]. Additionally, antibodies can be detected even 10 months after infection in unvaccinated individuals [13]. This may be the basis for creating future recommendations regarding vaccination of convalescents. The decision about the number of doses of the primary vaccination may depend either on the history of previous infection confirmed by positive PCR or antibody tests or antibody levels before the first vaccine dose. Before changing the current vaccination rules, this topic requires further investigation.

This study was conducted while Alpha, Beta and Delta variants were circulating and causing most infections. There are some scientific reports about convalescents benefiting from just one booster shot in terms of protection against Wuhan D614G, Delta and Omicron [19]. These findings are based on an in vitro study of 66 convalescents; therefore, the topic requires further exploration and real-world case examination.

There are no data about the effectiveness and the duration of vaccine-induced antibody responses among previously seropositive and seronegative individuals.

There is not yet information regarding how long after infection one shot would be sufficient. It is probably the same as in current recommendations concerning booster shots for everyone who received the primary series.

Nevertheless, the level of protection in convalescents may be different and dependent on the severity of previous SARS-CoV-2 infection [20,21]. Therefore, mass vaccination would be safer, and giving everyone the full primary series followed by a booster dose to avoid insufficient protection and appearance of viral variants was a simple solution to the emerging health crisis [22].

However, in some countries, results indicating high anti-SARS-CoV-2 antibody levels in convalescents led to temporary changes in the vaccination schedule and the use of a single dose of two-dose vaccine in this group of patients.

Another fact worth focusing on is that 61 out of 162 patients from group I presented lowering antibody results after the second dose compared to the first dose. This phenomenon may be explained by a few factors [23]. First of all, the timing of the sample (4–5 weeks after 2nd dose) may not be optimum for showing increased antibody production. Furthermore, all discussed patients presented substantially higher antibody titers after the first dose, and these titers could block produced SARS-CoV-2 antigens and could weaken the effectiveness of antigen-presenting cells, which would result in a limited immunological response after the second dose. Marie I. Savanovic suggested that this may also be a reaction to polyethylene glycol used by Pfizer, but in our study only 39 patients received the Pfizer vaccine (16 Moderna and 6 Astra Zeneca). Despite lower antibody levels, the results observed after second doses can still be considered high.

Although there is not much information about lower antibody levels after the second dose compared to the first dose, there are plenty of data considering prolonging intervals between two doses [24]. This shall result not only in a stronger immunological response of immunocompetent individuals, but it may also lower the number of severe adverse events, such as heart inflammation seen in some young men [25]. A total of 37.7% of convalescents did not benefit from the second dose, and an extended interval between two doses can help reduce severe side effects, which is an important factor in the discussion, especially with people skeptical about vaccinations. The individualization of vaccination schemes for convalescents in worldwide terms is possible just like in cases of immunocompromised patients who now receive three doses in the primary series and could be received with understanding and supporting by the general public.

Among our participants, 100% developed antibodies after the second/last dose, but four of them did not respond to the first dose (1.4% from 286 individuals analyzed after the first dose). These findings are consistent with the data obtained in other studies. Gareth Iacobucci reported that 96.42% out of 8517 patients in England and Wales developed antibodies 28 to 34 days after their first dose [26].

All types of vaccines studied in this project led to high antibody titers after a full vaccination schedule. Moderna showed up as the most effective in terms of highest antibody levels, although both mRNA vaccines are proven to be similar in their results [27]. Our findings regarding mild side effects most commonly appearing after Moderna shots are consistent with the available data [28]. This is further proof that mRNA technology is a significant and useful achievement of modern science. It is also worth mentioning that the Moderna vaccine is easier to transport and store than the one produced by Pfizer. Although patients in our study received four types of vaccines, we do not want to state similar conclusions about Astra Zeneca and Johnson&Johnson vaccines since sample sizes in these cases were small.

In the course of analyzing the acquired data, a significant correlation was found between some side effects and higher antibody levels after each dose, such as fever, weakness, pain at the injection site and muscle and joint pain. From the symptoms mentioned above, only fever in convalescents was observed to be relevant after the first dose. For both group I and II, a positive correlation was found between these symptoms reported after the second dose and the antibody titers. William Schaffner, M.D., stated in an interview for Medical News Today that there is no direct correlation between side effects and protection [29]. This does not mean that this stands in contradiction with our findings that focus on antibody levels rather than on protection levels against infection.

### 4.1. Limitations

We assessed only the humoral immune response in terms of IgG against SARS-CoV-2, aware that it is only a part of the immune response, but very easy to test and cheap. Based on the producer’s information, the test can to some extent detect neutralizing antibodies. The convalescent and naïve patient groups consisted of different numbers of patients, and only a few participants chose the Johnson&Johnson vaccine, which made it difficult to compare with other vaccines.

### 4.2. Conclusions

All four types of vaccines were effective, and they induced serological responses in immunocompetent patients, both in convalescents and naive patients. The best results came from the Moderna vaccine followed by Pfizer, but both mRNAs worked as expected.

We observed that in the case of naive patients, antibody levels increased after every dose. In the convalescents group with baseline level > 0 IU/mL, the increase of antibody levels after the first dose was significantly higher, but the baseline level had no significance regarding results after the second dose.

We compared results obtained from the convalescents group after the first dose and naive group after the second dose, and we conclude that the convalescents had been better immunologically stimulated.

A total of 37.7% of convalescents showed a decrease of antibody titers after the second dose compared to the first dose. It is worth emphasizing that in this subgroup, we had patients receiving Pfizer, Moderna and Astra Zeneca vaccines; therefore, the phenomenon was not dependent on the manufacturer. These are important discoveries which should lead to changes in the prime series of SARS-CoV-2 vaccinations for recent convalescents, especially young people with high antibody levels (e.g., >2080 IU/mL, which is a standard maximum level shown by commercial laboratories). They can benefit from prolonging the intervals between first and second doses in terms of immunological responses and additionally by lowering the risk of rare but severe side effects. We encourage elderly people and those with low antibody levels after past infections not to hesitate and to receive both vaccine doses in the recommended time.

Among the most commonly reported side effects, we found pain at the injection site, weakness, fever and muscle and skeletal pain. In the naive group, the symptoms were more intensified after the second dose, whereas in the convalescents group, they were often observed after the first dose. People receiving the Moderna vaccine more often reported adverse effects in the questionnaire and they also presented higher antibody levels. We are not keen to make similar conclusions regarding Astra Zeneca and Johnson&Johnson vaccines since the groups of patients receiving vector vaccines in our study were relatively small.

## Figures and Tables

**Figure 1 vaccines-10-01042-f001:**
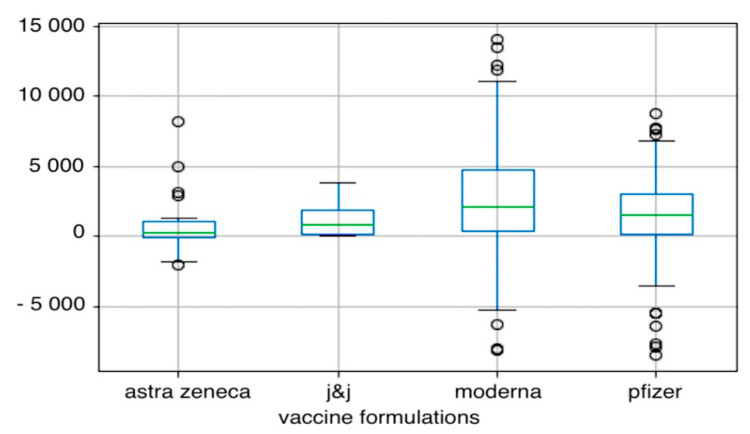
The figure shows the difference in antibody levels after the administration of the second dose of the individual vaccine formulations [IU/mL]. J&J-Johnson&Johnson. The main body of the boxplot shows the quartiles, horizontal lines in the middle of each box are medians, whiskers is the vertical lines extending to the most extreme, non-outlier data points and fliers represent data that extend beyond the whiskers.

**Figure 2 vaccines-10-01042-f002:**
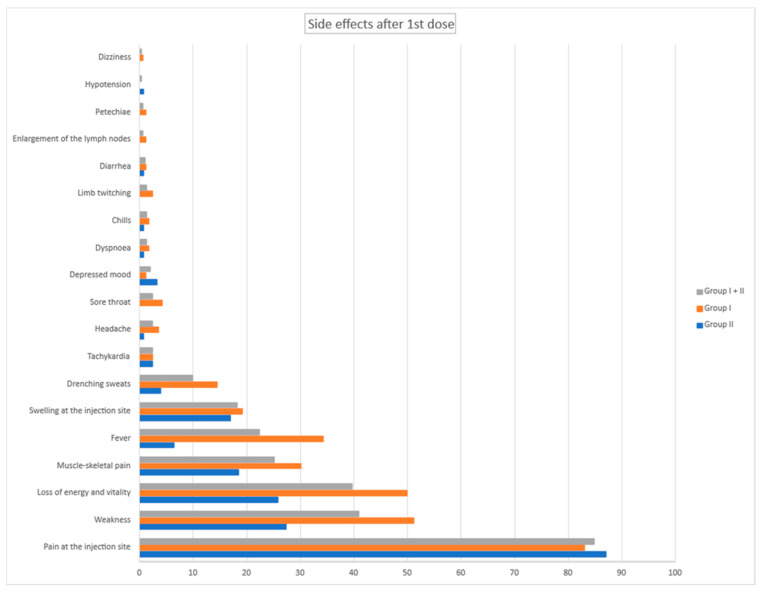
Side effects after 1st dose for group I and group II in %.

**Figure 3 vaccines-10-01042-f003:**
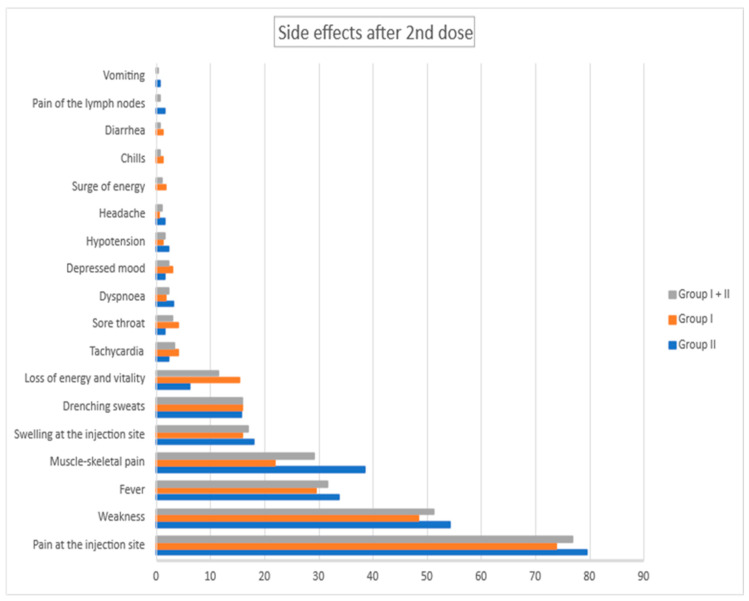
Side effects after 2nd dose for group I and group II in %.

**Figure 4 vaccines-10-01042-f004:**
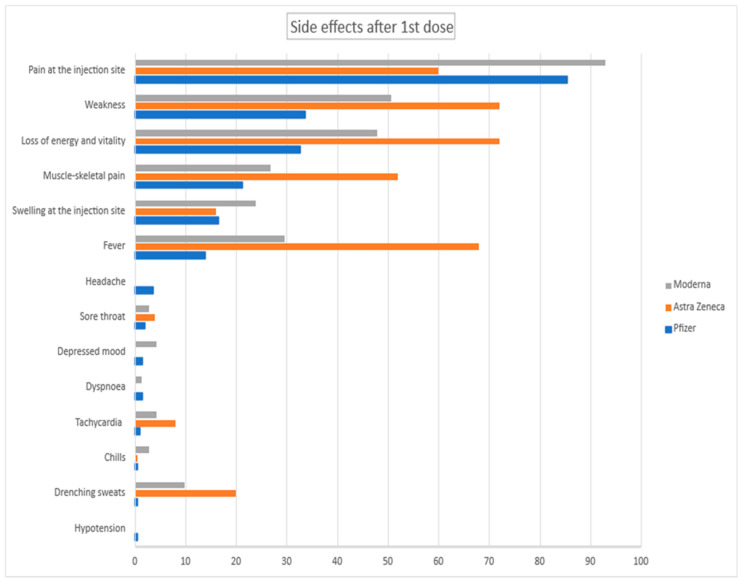
Side effects after 1st dose for specific vaccines in %.

**Figure 5 vaccines-10-01042-f005:**
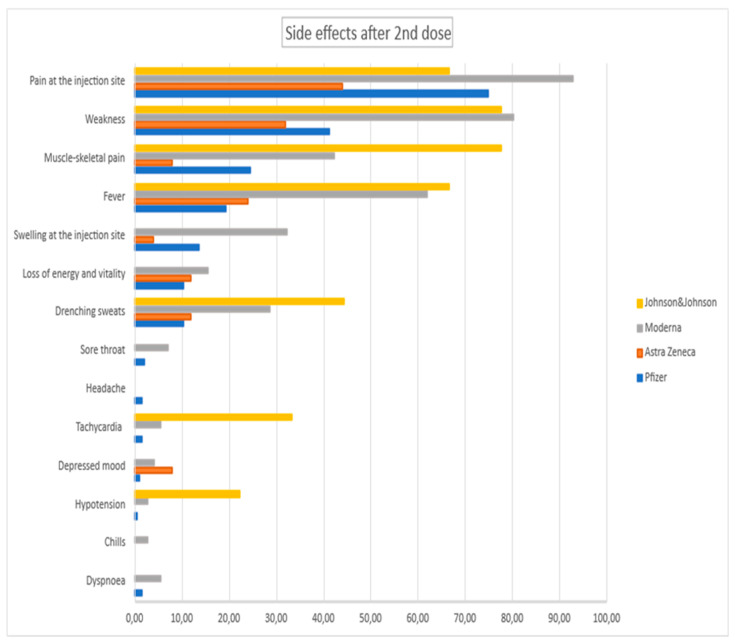
Side effects after 2nd dose for specific vaccines in %.

**Table 1 vaccines-10-01042-t001:** Characteristics of the groups.

Parameters		Group I (n = 171)	Group II (n = 127)	*p* Value
Age	Mean ± SD	44.0 ± 9.4	42.6 ± 7.2	0.177
	Me [IQR]	44.0 [39.0–49.0]	43.0 [39.0–47.0]	
Sex	female	101 (57.81%)	74 (58.27%)	0.828
	male	70 (42.19%)	53 (41.73%)	
Anti-SARS-CoV-2 antibodies level [IU/mL]	Before vaccination (Mean ± SD)	190.3 ± 328.4	0	<0.001
	Me [IQR]	105.6 [38.4–198.4]	0	
SARS-CoV-2 infection in the past	yes	150	0	
	no	21	127	
Vaccine type for D2	Pfizer	104	87	
	Moderna	37	33	
	Astra Zeneca	21	4	
	Johnson&Johnson	6	3	

SD— standard deviation; Me—median; IQR—interquartile range.

**Table 2 vaccines-10-01042-t002:** Antibody levels after the first dose of anti-SARS-CoV-2 vaccine in convalescents [IU/mL].

Group	Mean	N	SD	MIN	MAX	Me [IQR]
BASELINE = 0	3684.8	23	2423.4	0.0	7936.0	3699.2 [1248.0; 5239.5]
BASELINE > 0	5826.5	138	4586.1	0.0	25,600.0	4862.3 [2944.0; 6336.0]

SD—standard deviation; Me—median; IQR—interquartile range.

**Table 3 vaccines-10-01042-t003:** Anti-SARS-CoV-2 IgG antibody levels at each stage of the study.

	Anti-SARS-CoV-2 IgG Antibodies Levels [IU/mL]					
	D0-Baseline		D1—After 1st Dose		D2—After 2nd/Last Dose	
	Mean ± SD	Me [IQR]	Mean ± SD	Me [IQR]	Mean ± SD	Me [IQR]
Group I	190.3 ± 328.4	105.6 [38.4; 198.4]	5501.5 ± 4380.0	4736.0 [2676.9; 6144.0]	5523.7 ± 4016.4	4560.0 [3040.0; 7232.0]
Group II	0.00 ± 0.00	0.0 [0.0; 0.0]	751.9 ± 897.7	451.2 [217.6; 915.2]	3625.6 ± 2568.9	3056.0 [2048,0; 4512.0]
*p*-value	0.00000		0.00000		0.00000	

SD—standard deviation; Me—median; IQR—interquartile range.

**Table 4 vaccines-10-01042-t004:** Comparison of two subgroups from group I—patients who presented lower antibodies level after second dose of vaccine (A) vs. patients who presented constant increase of antibodies level (B).

	Group I	Mean	N	SD	MIN	MAX	Me [IQR]
Age (*p* = 0.0418) [years]	A	42.3	61	9.7	21.0	68.0	43.0 [38.0; 46.0]
	B	45.4	101	9.2	24.0	69.0	45.0 [40.0; 52.0]
Baseline level (*p* = 0.939) [IU/mL]	A	189.4	61	287.0	0.0	1600.0	120.0 [46.4; 188.8]
	B	185.4	99	338.2	0.0	2688.2	83.2 [38.3; 204.8]
after 1st dose (*p* = 0.0001) [IU/mL]	A	6983.1	61	4698.9	787.2	21,800.0	5376.0 [4000.0; 8256.0]
	B	4606.9	101	3960.2	0.0	25,600.0	3840.0 [2080.0; 5587.2]
After 2nd dose (0.0542) [IU/mL]	A	4811.3	61	3192.5	533.0	19,840.0	3968.0 [3008.0; 6048.0]
	B	4879.7	101	4397.9	486.4	28,200.0	4992.0 [3132.0; 7936.0]

SD—standard deviation; Me—median; IQR—interquartile range.

**Table 5 vaccines-10-01042-t005:** Antibodies levels after particular SARS-CoV-2 vaccines.

	Astra Zeneca		Johnson&Johnson		Moderna		Pfizer	
Antibodies level	Mean ± SD	Me [IQR]	Mean ± SD	Me [IQR]	Mean ± SD	Me [IQR]	Mean ± SD	Me [IQR]
baseline	283.5 ± 608.3	88.0 [33.6; 176.8]	219.4 ± 502.1	28.8 [0.0; 134.4]	101.9 ± 245.1	0.0 [0.0; 96.0]	84.7 ± 163.5	0.0 [0.0; 122.4]
After 1st dose	1800.0 ± 1611.1	1376.0 [787.2; 2768.3]			4851.6 ± 5236.0	3232.0 [1014.4; 6080.0]	3228.3 ± 3777.9	2256.0 [358.4; 5024.0]
After 2nd/last dose	2841.8 ± 2398.6	1920.0 [1074.0; 3904.0]	1337.4 ± 1448.6	844.8 [140.8; 1920.0]	4733.1 ± 4733.1	5316.0 [3904.0; 8540.0]	4369.1 ± 2893.1	3360.0 [2464.0; 5456.0]

SD—standard deviation; Me—median; IQR—interquartile range.

**Table 6 vaccines-10-01042-t006:** Antibodies levels after particular SARS-CoV-2 vaccines.

	Astra Zeneca	Johnson& Johnson	Moderna	Pfizer
Astra Zeneca	-	-	-	-
Johnson& Johnson	U = 80.0 *p*-value = 0.1675	-	-	-
Moderna	U = 547.5 *p*-value = 0.0062	U = 250.0 *p*-value = 0.1236	-	-
Pfizer	U = 1592.5 *p*-value = 0.0135	U = 780.5 *p*-value = 0.3049	U = 6024.0 *p*-value = 0.0342	-

SD—standard deviation; Me—median; IQR—interquartile range.

## Data Availability

The full data presented in this study are available on request from the corresponding author.

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
