# Peer review of "How Humoral Response and Side Effects Depend on the Type of Vaccine and Past SARS-CoV-2 Infection"

_vaccines, 2022, doi:10.3390/vaccines10071042_

Round 1

Reviewer 1 Report

Thank you for inviting me to review this manuscript. The manuscript reports on a study investigating how humoral response and side effects depend on the type of vaccine and past SARS-Cov-2 infection. In general, the paper reads well and merits attention. However, I feel there are a few problems that should be improved.

1. The introduction seems very short. More in-depth information is needed to be included.

2. The sampling method and sample size are not sufficiently explained. Please indicate the sample size calculation and how you recruited the sample more clearly.

3. I recommend subtitle Methods use The procedure.

4. Please clearly indicate what was or what was the study outcomes. Although you explain this, the presentation is a bit confusing.

5. After limitations I think you need to add a Conclusion. Currently, this is missing.

Author Response

1. The introduction seems very short. More in-depth information is needed to be included.

We put additional information and elaborate the main topic in the introduction.

2. The sampling method and sample size are not sufficiently explained. Please indicate the sample size calculation and how you recruited the sample more clearly.

We explained in detail how patients were recruited and how we established the final groups' sizes.

3. I recommend subtitle Methods use The procedure.

In the subtitle word 'methods' was changed to 'the procedure'

4. Please clearly indicate what was or what was the study outcomes. Although you explain this, the presentation is a bit confusing.

We indicated our findings and the study outcomes in the conclusions.

5. After limitations I think you need to add a Conclusion. Currently, this is missing.

We added conclusions paragraph which should make the paper and the outcomes more clear.

Reviewer 2 Report

My main concern is about the test to quantify abs in volunteers vaccinated with Moderna and Pfizer versus Astra Seneca and Jhonson and Jhonson. mRNA vaccines developed by Pfizer-BioNTech and Moderna were developed to generate a protein (Spike) that itself safely prompts an immune response. On the other hand, Viral vector (adenovirus) vaccines developed by Astra-Zeneca and Johnson & Johnson, use a virus that has been genetically engineered to produce coronavirus proteins (spike protein) to safely generate an immune response. I wonder if the Antibodies levels after SARS-CoV-2 vaccines (in all grops/all determinations) or in convalescent patients were exactly quantified/determined in all groups?. Do you consider that the humoral response (Antibodies titers) is well quantified/established by this method? Authors must to explain or speculate about this, also in the discussion.

Author Response

My main concern is about the test to quantify abs in volunteers vaccinated with Moderna and Pfizer versus Astra Seneca and Jhonson and Jhonson. mRNA vaccines developed by Pfizer-BioNTech and Moderna were developed to generate a protein (Spike) that itself safely prompts an immune response. On the other hand, Viral vector (adenovirus) vaccines developed by Astra-Zeneca and Johnson & Johnson, use a virus that has been genetically engineered to produce coronavirus proteins (spike protein) to safely generate an immune response. I wonder if the Antibodies levels after SARS-CoV-2 vaccines (in all grops/all determinations) or in convalescent patients were exactly quantified/determined in all groups?. Do you consider that the humoral response (Antibodies titers) is well quantified/established by this method? Authors must to explain or speculate about this, also in the discussion.

We carefully established the antibody titers of the participants. Serological testing has been used to verify vaccination effectiveness since many years e.g. in case of anti-HBV vaccine and we believe it is an adequate method to evaluate SARS-CoV-2 vaccines. We added more information about the topic in the manuscript. We would like to underline that we do not feel that our sample sizes for Astra Zeneca and Johnson&Johnson vaccines are big enough to compare the results with the mRNA vaccines. 

Reviewer 3 Report

 Regarding the efficacy of the vaccine against Covid-19 in Poland, a patient group was set up, and each company's vaccine was reported in detail regarding the measurement of anti-SARS-CoV-2 antibody levels before and after vaccination and the interview regarding side effects. It contains data that is useful to the reader, but I encourage you to accept this paper after making corrections and additions to the following two points.

1:  I felt that the thoughts and opinions of the authors were not reflected very much in the summary and discussion. I hope that you will not only enumerate the results of the analysis, but also discuss what can be inferred from this fact.

2:  It should be clearer what the conclusions are about the humoral response and side effects of vaccination against SARS-CoV-2 infection. It is unclear and difficult to understand.

Author Response

1:  I felt that the thoughts and opinions of the authors were not reflected very much in the summary and discussion. I hope that you will not only enumerate the results of the analysis, but also discuss what can be inferred from this fact.

We added our opinions and interpreations of the results into the manuscript.

2:  It should be clearer what the conclusions are about the humoral response and side effects of vaccination against SARS-CoV-2 infection. It is unclear and difficult to understand.

We created a separate paragraph with conclusions in the end of the manuscript to sum up the results and meaning of our findings.

Round 2

Reviewer 1 Report

Thank you for inviting me to review this revised version. I think the manuscript improved and the authors responded to my earlier comments. No further comments.